# Sodium Nitrite Intoxication and Death: Summarizing Evidence to Facilitate Diagnosis

**DOI:** 10.3390/ijerph192113996

**Published:** 2022-10-27

**Authors:** Martina Padovano, Mariarosaria Aromatario, Stefano D’Errico, Monica Concato, Federico Manetti, Maria Chiara David, Matteo Scopetti, Paola Frati, Vittorio Fineschi

**Affiliations:** 1Department of Anatomical, Histological, Forensic and Orthopedic Sciences, Sapienza University of Rome, 00161 Rome, Italy; 2Unit of Risk Management, Quality, and Accreditation, Sant’Andrea University Hospital, 00189 Rome, Italy; 3Department of Medicine, Surgery and Health, University of Trieste, 34137 Trieste, Italy; 4Department of Public Security, Health Central Directorate, Research Center and Forensic Toxicology Laboratory, Ministry of the Interior, 00184 Rome, Italy; 5Department of Medical Surgical Sciences and Translational Medicine, Sapienza University of Rome, 00189 Rome, Italy

**Keywords:** sodium nitrite, fatal methemoglobinemia, post-mortem investigations, suicide, forensic pathology, autopsy findings, post-mortem toxicology

## Abstract

Background: Over the years, forensic pathology has registered the spread of new methods of suicide, such as the ingestion of sodium nitrite. Sodium nitrite causes increased methemoglobin, resulting in systemic hypoxia, metabolic acidosis, and cyanosis. Since sodium nitrite is a preservative, the ingestion of foods containing an excessive amount of this substance can also cause acute intoxication up to death. The present review is aimed at guiding health professionals in the identification and management of sodium-nitrite-related intoxications and deaths. Methods: A systematic literature search was carried out on PubMed by following the PRISMA statement’s criteria. A total of 35 studies with 132 cases were enrolled, and the data were cataloged in Microsoft Excel. To establish the causal correlation between sodium nitrite ingestion and death, the Naranjo Adverse Drug Reaction Probability Scale was used. Results: In addition to the small number of cases that have currently been published, the study demonstrated that there was a general methodological discrepancy in the diagnostic process. However, some interesting results have emerged, especially in post-mortem diagnostics. Conclusion: Sodium-nitrite-related deaths represent a challenge for forensic pathologists; therefore, it is important to promptly recognize the essential features and perform the necessary and unrepeatable examinations for the correct diagnosis of the cause of death.

## 1. Introduction

Nowadays, suicide continues to represent a burden that affects the entire world population. According to the World Health Organization (WHO), suicide-related deaths are estimated to be around 703,000 annually, and 58% involve people under 50; moreover, the data show that 77% of suicide cases are recorded in low- and middle-income countries [1]. Data published by the Centers for Disease Control and Prevention (CDC) in 2022, which were in relation to the US population for the twenty years from 2000 to 2020, showed that the suicide rate increased by 30%, with a peak that was reached in 2018 and with a 3–4 times higher rate in males [2]. In fact, individuals of the male and female genders have different approaches to this practice; specifically, men are more frequently involved in effective suicidal events, contrary to women, who more frequently only attempt suicide. Furthermore, one difference consists in the propensity of males to choose methodologies that are characterized by greater lethality and violence [3]; hanging represents the most frequent method used by men, while drug abuse is the method used most frequently by women [4,5]. There are conflicting data when considering different countries. For example, in countries where the possession of weapons is widespread, such as the USA, the main method of suicide in men is by firearms [6]. Acute intoxication with exogenous substances can occur through the adoption of different modalities, such as, primarily, ingestion, inhalation, and poisoning by chemical substances, including pesticides [7]. The advent of the internet, as well as the evolution of society and life habits, has determined a heterogenization of suicide methods over the years. The particular cases previously reported in the literature are progressively increasing, thus allowing one to highlight how new methods of suicide suddenly spread.

Among these, the acute ingestion of sodium nitrite is becoming increasingly important. Sodium nitrite is an inorganic compound that, at environmental temperature, has a crystalline, odorless, yellowish-white, water-soluble, and hygroscopic appearance. This compound is characterized by its easy availability and has found applications in several fields. It is widely prescribed in medicine, where it is mainly administered as an antidote to cyanide poisoning [8], while, in the food industry, it is used as a preservative (labeled with the code “E250”) in meat products to inhibit bacterial growth and prevent deterioration, as well as to improve organoleptic characteristics (appearance, color, and taste) through a reaction with myoglobin [9]. Given its dose-dependent toxicity, over the years, international governmental bodies have regulated the use of this preservative to protect public health and food safety [10,11,12]. Despite this, the same restrictions have not been applied in the context of trade. Still today, this substance is easily available, even online. Its easy availability and affordable cost have allowed its progressive and sudden spread as a means of suicide, even among young people of both genders.

Sodium nitrite results in an increase in the concentration of methemoglobin (MHb), which normally constitutes only 1–2% of the total circulating hemoglobin. Hemoglobin is a tetrameric metalloprotein with four subunits, each of which is bound to a heme group capable of binding an Fe^2+^/Fe^3+^ ion. Normally, heme binds ferrous iron (Fe^2+^), which is capable of transporting the oxygen molecule so that it can be released into the tissue. Sometimes, such as through the action of sodium nitrite, ferrous iron is oxidized to ferric iron (Fe^3+^), thus losing its ability to bind oxygen and resulting in a greater affinity for oxygen in the remaining Fe^2+^ heme groups that are present in the same hemoglobin structure. Thus, hemoglobin is transformed into MHb, which has a lower capacity to transport oxygen and release it into the tissue [13]. In fact, the allosteric modifications of MHb, in addition to allowing bonds with a smaller number of oxygen molecules, result in the establishment of a stronger bond, which is the reason for the difficulty of the release of the oxygen into the tissue; therefore, clinically, the percentage of residual oxyhemoglobin constitutes only the theoretical oxygen transport capacity, as the effective peripheral release capacity must be considered. The reduction in aerobic cellular metabolism resulting from systemic hypoxia due to MHb results in heterogeneous conditions that range from metabolic acidosis to cyanosis and death [14].

The present systematic review, which was conducted according to the PRISMA 2020 guidelines, aspires to frame the main features of deaths due to sodium nitrite intake; specifically, this research aims to summarize the current evidence derived from the small number of cases reported in the literature. As a secondary objective, this study focuses on post-mortem assessments to provide support to healthcare professionals who are involved in the investigation of sodium-nitrite-related deaths.

## 2. Materials and Methods

### 2.1. Search Strategy and Study Selection

The present review involved a systematic literature search carried out up to 21 June, 2022 on PubMed while following the PRISMA (Preferred Reporting Items for Systematic Reviews and Meta-Analyses) statement’s criteria [15,16], without time limits (Figure 1). In order to broaden the number of studies included in the search, MeSH terms, Boolean operators, and free-text terms were used to carry out the review. Articles focusing on the effects of sodium nitrite ingestion were initially searched using the terms “sodium nitrite AND ((intoxication) OR (death) OR (suicide) OR (ingestion))” in the title, abstract, and keywords. The study design included case reports and case series. No unpublished or gray literature was consulted. A total of 56 studies relevant to the present review were identified on PubMed; of these, 12 were excluded due to unavailability and 11 were excluded because they were not in English. The analysis of the references during the full-text reading allowed the inclusion of a further 6 studies. After evaluating the abstracts and the full texts, 4 articles were excluded because they were not relevant to the research topic. Therefore, the present review involved the inclusion of a total of 35 studies, which were divided into 15 case reports and 20 case series. The data in each included study were extracted by using Microsoft Excel spreadsheets; these data included information on the authors, journal, year of publication, country, sample size, gender, age, comorbidities, method of assumption, intentionality, any other substances taken, symptoms and times of manifestation, hospitalization, mortality, scene investigations, post-mortem assessments, and diagnosis of death.

### 2.2. Causality Assessment

This study considered a causality assessment by applying the Naranjo Adverse Drug Reaction Probability Scale [17] to validate the evidence provided by the review. This scoring algorithm, which is used in the context of pharmacovigilance, was applied to attribute to sodium nitrite a scientifically objective causal correlation with intoxication and death. The scale consists of 10 questions with 3 possible answers, each of which is assigned a score that allows the attribution of the probability of such a causal correlation in each case. The scores are between −4 and +12, and they permit the following division of adverse drug reactions based on the probability of their existence:−definite (score ≥ 9);−probable (score between 5 and 8);−possible (score between 2 and 4);−doubtful (score < 2).

## 3. Results

### 3.1. General Features

The present systematic review included a total of 35 papers, divided into 15 case reports and 20 case series (Table 1); 11 studies were carried out in America, as well as 11 in Asia, nine in Europe, three in Australia, and one in Africa. Overall, a total of 132 subjects were included (75 males, 34 females, and 23 of undefined gender); ages were available for 97 cases and ranged from 2 to 76 years, with an average of 35.6 years (Figure 2). In 53 of the 132 cases, the ingestion of sodium nitrite occurred for suicidal purposes, while in the remaining 79 cases, it occurred accidentally. A total of 21 of the 53 suicides were hospitalized, but 12 of these died anyway. However, 73 of the 79 individuals with accidental ingestions accessed the emergency room, and of these, only six died during hospitalization.

### 3.2. Medical History

In 95 cases, it was possible to collect information relating to the symptoms (Figure 3). The symptoms that were most commonly presented were cyanosis of the face and/or extremities (64; 48.49%), breathing abnormalities (64; 48.49%), altered level of consciousness (57; 43.18%), dizziness (30; 22.73%), vomiting (28; 21.21%), nausea (22; 16.67%), pain (headache: 18, 13.64%; abdominal pain: 1, 0.76%, and chest pain: 1, 0.76%), and tachycardia (19; 14.39%).

A psychiatric anamnesis was collected in only 10 cases of ingestion for suicidal purposes. The psychiatric histories included depression (7; 70%), previous suicide attempts (6; 60%), bipolar disorder (2; 20%), schizophrenia (1; 10%), paranoia (1; 10%), previous self-harm attempts (1; 10%), and post-traumatic stress disorder (1; 10%).

### 3.3. Post-Mortem Investigations

Out of the total number of cases, death occurred in 55 of the subjects (41.67%) (Table 2); specifically, it occurred in the cases of 40 males (72.73%), 13 females (23.64%), and two subjects without information about their gender (3.63%) (Figure 4). Concerning the reason for the consumption, in 44 cases (80%), suicidal intent was found, while in 11 cases (20%), the event was accidental. Of the 55 cases, the majority (37; 67.27%) were found dead and, therefore, not hospitalized.

In 24 cases, an investigation of the scene made it possible to identify particular objects near the body or in adjacent rooms. Specifically, the presence of at least one open container containing crystalline, odorless, and yellowish-white-colored powder, possibly bearing the label “Sodium nitrite”, was identified; usually, spoons, bottles, and glasses soiled with a whitish substance were also present. Sometimes, suicide notes and other medications were found.

In 39 cases, a post-mortem examination was performed. On external inspection, lividity varied in color from bluish red to mottled purple–gray; intense cyanosis of the face and extremities (up to gunmetal gray) was also highlighted. The autopsy revealed the presence of extremely thin and chocolate-colored blood and brown-colored organs—mainly the heart, kidneys, and liver—as well as whitish and paste-like residues in the gastric lumen; sometimes, these findings were associated with pulmonary and cerebral edema.

Microscopic examinations were carried out in only five cases. Histopathology revealed alterations related to comorbidities (myocardial fibrosis, coronary artery disease), signs of hypoxia (early myocardial ischemia), and non-specific findings (pulmonary edema).

Regarding post-mortem toxicology, methemoglobin (MHb) was studied in the blood, gastric content, and cerebrospinal and pericardial fluid (Table 3). The investigation of blood samples allowed the detection of concentrations between 9.87% and 83.4% in 26 cases; it must be specified that in 3 of the 26 cases, the investigations gave a positive result, in another three cases, the value exceeded the laboratory’s upper limit of detection, and in two cases, the results were not available. In 1 of the 26 cases, the dosage was studied in both a blood sample taken during the cadaveric inspection (MHb: 33%) and in two samples carried out during the autopsy (one of heart blood and one of peripheral blood; MHb: 26%); the results of tests in gastric content and cerebrospinal and pericardial fluid were not available.

A search for nitrite was conducted in the blood, vitreous humor, gastric and bowel content, urine, cerebrospinal and pericardial fluid, liver, kidney, and costal cartilage. Blood samples taken during the autopsy were tested in 16 cases; in detail, the concentrations in heart blood ranged from 1.3 to 170 mg/L, while in peripheral blood, they varied from 1.1 to 298 mg/L; in two cases, the combined investigations in heart and peripheral blood samples gave negative results, but the correlation between death and sodium nitrite was, however, confirmed by other evidence. Vitreous humor was tested in one case (57.7 mg/L), gastric content in 15 cases (range: 6.8–16,000 mg/L), and urine in one case (24.6 mg/L); in one case, the investigation of nitrite was performed in cerebrospinal fluid (negative) and pericardial fluid (181 mg/L). An investigation of nitrite was also carried out in one case on a liver and kidney mixture (0.003 mg/kg); in one case, the liver (0.3 mg/kg), kidney (3.6 mg/kg), and costal cartilage (3.4 mg/kg) were tested; in one case, the dosage of nitrite in bowel content was assessed (4.2 mg/L).

In one case, sodium nitrite was assessed in blood with a positive result. Gastric content was also tested for sodium nitrite in seven cases; in two cases, the concentrations were equal to 13,000 and 24,000 mg/L, in four cases, the test was positive, and in one case, it was negative.

The search for nitrate was performed on blood and on gastric and bowel content, as well as cerebrospinal and pericardial fluid. In 13 cases, autopsy samples of heart blood (range: 71.69–524.8 mg/L) and peripheral blood (range: 83.48–476.4 mg/L) were tested; in one case, a peripheral blood sample taken during the cadaveric inspection was examined, and it had a value of 220.6 mg/L. Nitrate was also detected in gastric content in two cases (5.0 mg/L, 137.8 mg/L), in bowel content in one case (0.8 mg/L), and in cerebrospinal fluid (50.5 mg/L) and pericardial fluid (91.7 mg/L) in one case.

Sodium (148 mmol/L) and chloride (124 mmol/L) were examined in vitreous humor in one case.

Substances of abuse and/or drugs were investigated in blood and urine in 25 cases; the tests were positive in 13 cases and negative in nine cases; the results were not available in three cases.

### 3.4. Causality Assessment

The attribution of a numerical value according to the Naranjo Adverse Drug Reaction Probability Scale made it possible to detect a score of 10 in 69 cases (36 survivors, 33 deceased), a score of 9 in 14 cases (11 survivors, 3 deceased), a score of 8 in 10 cases (9 survivors, 1 deceased), and a score of 7 in 39 cases (21 survivors, 18 deceased).

In consideration of the stratification of the categories and the scores obtained, the causal correlation between sodium nitrite ingestion and intoxication/death was definite in 83 cases (62.88%; 47 survivors, 36 deceased) and probable in 49 cases (37.12%; 30 survivors, 19 deceased) (Figure 5).

## 4. Discussion

The present systematic review aimed to analyze suicidal or unintentional ingestion of sodium nitrite; specifically, the features of the involved subjects (sex, age, and comorbidities), circumstantial data (manner of intake, intentionality, and other substances taken), symptoms, hospitalization, and post-mortem investigations (if any) were considered.

Particular attention was paid to the ingestion of sodium nitrite for suicidal purposes, a pervasive phenomenon that is afflicting public health worldwide. The prevention of the risk of suicide constitutes a challenge for suicidology, which tries daily to face ever-new anticonservative methodologies [53]. To prevent both suicidal behaviors and suicides, it is advisable to implement a multidisciplinary approach based on the involvement of different complementary professional figures in both the hospital and the territorial setting on the training of non-healthcare professionals. An aspect that is not to be underestimated is the involvement of the media, which should play an important role in suicide prevention campaigns in order to portray suicide as an act that should be avoided and not emulated, which is often, unfortunately, the case [54].

Nowadays, in parallel with societal change, the culture of suicide is also evolving in both its methods and its age groups. In fact, the increasing diffusion and the simplicity with which it is possible to access devices connected to the internet have allowed the spread of content to an ever-wider audience [55]. Currently, through dozens of social networks, it is possible to share one’s thoughts and contributions with the rest of the world, as well as, vice versa, to learn new knowledge from the latter. Sometimes, shared content is not characterized by social utility and legality. This is the case on many websites that list multiple methods of suicide and even try to induce mentally fragile subjects to partake in them [56]. Such free and often uncontrolled communication, which borders on illegality, causes extreme situations such as suicide to be ever closer and more realistically feasible. It is not rare to find cases in which ways to commit suicide are researched online [57,58]. From this perspective, information relating to the action and availability of substances such as sodium nitrite has been made known to the group of the population that uses the web the most—young people [59,60].

The evidence that arose from the 35 studies considered here permits the confirmation of this phenomenon; in fact, it emerged that 90.6% of the subjects who ingested sodium nitrite for suicidal purposes were aged between 15 and 39 years and, thus, belonged to the population that used the internet as a main source of information the most.

Given the spread of the phenomenon, it would be advisable, on the one hand, to implement tight checks of online content and, on the other hand, to tackle the health problem. More precisely, from the perspective of prevention and public health, it appears to be important to protect the categories that are at risk by limiting the diffusion of content alluding to the use of sodium nitrite for suicidal purposes.

Therefore, in hospitals, one important aspect is the training of health professionals—not necessarily psychiatrists—to quickly identify an acute intoxication and to implement the necessary treatments (intravenous methylene blue and life-support therapies) [61,62]. This concept is also supported by the data that emerged from the studies considered here. In fact, the results highlight how mortality is higher in non-hospitalized subjects due to the absence of adequate drug and life-support therapy.

Mortality was significantly higher in cases of voluntary intake than in cases of involuntary intake, which was certainly due to the greater quantities of sodium nitrite that were taken in the event of a suicide attempt. The intake of an excessive but non-lethal quantity of sodium nitrite guarantees a longer survival interval, with the possibility of accessing an emergency department and receiving the appropriate therapy [63]. Another aspect is linked to the condition of solitude in which suicidal acts are carried out. In fact, it is more frequent that the consumption of food containing an excessive quantity of sodium nitrite occurs in the presence of witnesses who are able to provide help and quickly transport the subject to the hospital.

The rapidity in causing death and the simple availability of sodium nitrite have allowed the increased diffusion of this substance as a means of suicide [64]. Therefore, to establish its role in provoking death, it is essential, firstly, to identify all cases, even if only suspicious, and, secondly, to carry out all of the necessary post-mortem investigations to make the correct diagnosis of death. According to the evidence obtained here, an important role in framing the cases is played by the investigation of the scene. An adequate observation of the environment in which the body was found allowed the identification of peculiar objects—specifically, the discovery of open plastic packages containing crystalline, odorless, and yellowish-white-colored powder, which possibly bore the label “Sodium Nitrite”, and spoons soiled with whitish material, as well as bottles and glasses with a deposit of the same substance on the bottom. Such items may be associated with a suicide note and other drugs. These characteristic findings should constitute the main elements to be sought in cases of suspected ingestion of sodium nitrite, or, in the absence of suspicion, should represent the main objects noticed during the investigation of a scene to direct further examinations. The centrality of the investigation of the scene needs to be affirmed considering the results. The currently available literature analyzed in the present study attested that an inspection was ordered in only 48.98% of cases. Such a frequency, although significant, precludes the possibility of immediately obtaining data that are capable of directing the diagnosis in many cases.

An autopsy must be accurate to highlight peculiarities [65] such as a change in hypostasis in the extremities and face, which can range from dark-bluish red to mottled purple–gray, which is associated with intense cyanosis, up to gunmetal gray, as well as fluid and chocolate-brown blood. In some cases, the organs—mainly the heart, kidneys, and liver—were also brown in color; drug residues were also detectable in the gastric content. Sometimes, these findings may be associated with non-specific signs that are frequent in deaths related to the intake of exogenous substances, such as pulmonary and cerebral edema. Although they were frequent in the cases observed, the external signs were not pathognomonic and were rarely able to condition the subsequent diagnostic approach. Nevertheless, caution and methodological rigor are recommended in the execution of an external examination in order to interpret the findings correctly and to plan the subsequent investigations fruitfully. Furthermore, autopsy is fundamental for the sampling of biological fluids and tissues in order to search for nitrite, nitrate (obtained from the reaction of nitrite with oxyhemoglobin in the blood) [66], and the methemoglobin (MHb) dosage.

The analysis of data regarding toxicology was very difficult due to the methodological discrepancies in the available studies. Despite the differences, the most commonly used matrices included blood (for the investigation of MHb, nitrite, and nitrate) and gastric content (for nitrite investigations). In a minority of cases, urine, cerebrospinal fluid, vitreous humor, pericardial fluid, and others (such as the liver, kidney, bowel content, and costal cartilage) were taken. Specifically, it emerged that the direct search for nitrites is not a routine examination and was carried out only in a limited percentage of cases. This evaluation can be performed on several biological matrices, but the most commonly used are blood (for the entry of the substance into circulation) and gastric contents (for the methods of intake). Furthermore, although they were always higher than the normal percentage (1–2%), the MHb values showed wide variability, with a range from 9.87% to 83.4%. Although values around 70% are usually fatal [67], concentrations that are not too far from the normal range do not allow the exclusion of sodium nitrite as the cause of death, especially in cases in which comorbidities coexist. This statement is supported by studies in the literature that show that, in the clinical setting, the percentage of residual oxyhemoglobin represents only a theoretical capacity for oxygen transport, as it is necessary to take the effective tissue release capacity into account. The latter is strongly influenced by allosteric modifications of MHb, which are at the basis of a lower ability to transport oxygen, as well as a lower ability to release it. However, it is not possible to unambiguously determine the MHb concentration, especially in post-mortem analyses. In fact, these values may artificially be shown to be, on the one hand, increased after repeated freezing and thawing cycles and the lack of use of preservatives and, on the other hand, reduced due to the lack of prevention of the intracellular enzymatic reduction of MHb to hemoglobin and putrefactive hemolysis [68,69,70,71]. Therefore, the general orientation of forensic pathologists is to consider a causal correlation between the ingestion of sodium nitrite and death even for lower MHb values, as long as this association is substantiated by further objectivity [72].

Regarding accidental ingestion, despite the reduced frequency, it is important to focus on some epidemiological and preventive aspects. In the context of accidentality, most cases of intoxication can be traced back to the uncontrolled production of homemade foods. In such circumstances, the involvement of groups of people with heterogeneous clinical manifestations is not uncommon. From the clinical point of view, the manifestations are frequently mild, as no significant quantities of the additive are taken. In cases of survival, access to hospital care is extremely frequent. However, the index of clinical suspicion can sometimes be very low under consideration of the non-specificity of symptoms (nausea, vomiting, dyspnea, altered state of consciousness, and tachycardia). In fatal cases, the symptoms are mainly characterized by cyanosis, respiratory and hemodynamic instability, and alteration of consciousness. Even so, about half of the subjects die before reaching the hospital, which is probably due to the ingestion of high doses of sodium nitrite and a coincidence of chronic diseases. The scenario outlined here emphasizes the role of anamnestic collection in identifying patients to be sent for targeted diagnostic tests, including toxicological ones.

Given the above, the evidence that is currently available permits the presentation of an outline of practical advice for health professionals and forensic pathologists who are involved in cases of intoxication or death from sodium nitrite (Figure 6).

Finally, a problem of considerable importance is represented by the attribution of the causal relationship between the intake of the substance and intoxication or death. This task is particularly demanding, especially in cases in which the evidence highlighted cannot be univocally interpreted [73]. This gap can largely be filled by answering some questions about the phenomenology of the reaction to sodium nitrite ingestion.

## 5. Conclusions

Intoxication and death due to the ingestion of sodium nitrite represent a challenge for forensic pathologists. In fact, whether it is suicidal or unintentional ingestion, physicians must develop the necessary knowledge to suspect this methemoglobinemia to carry out necessary and unrepeatable tests for the correct diagnosis [74]. Future research and the sharing of evidence will guarantee the implementation of knowledge, as well as an ever more immediate identification of suspected cases.

## Figures and Tables

**Figure 1 ijerph-19-13996-f001:**
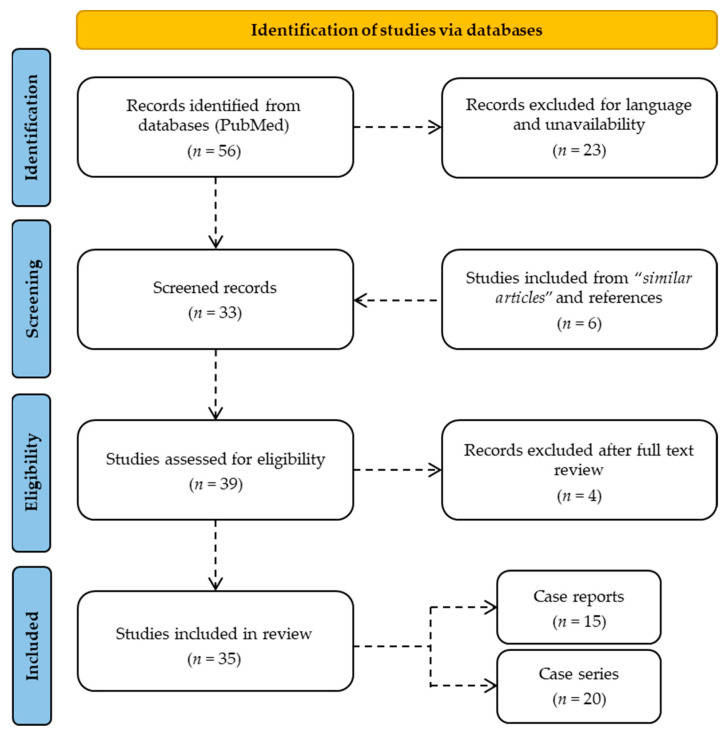
Flow diagram of the design of the study in accordance with the PRISMA 2020 guidelines.

**Figure 2 ijerph-19-13996-f002:**
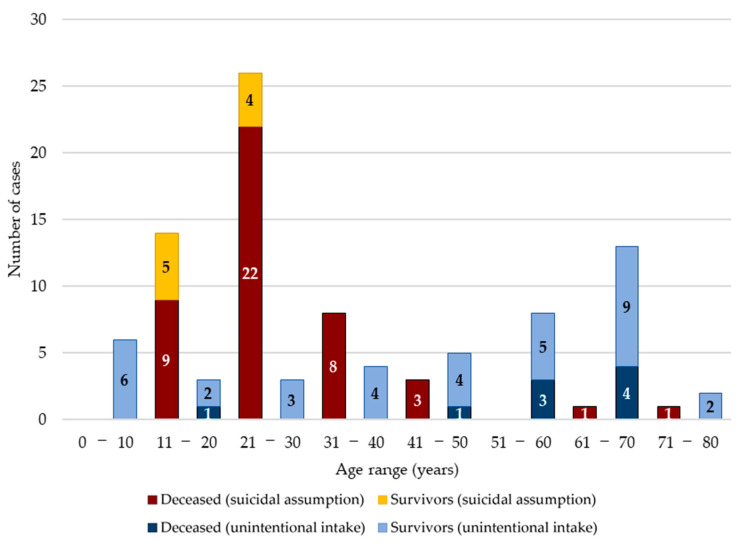
Stratifications by age range; there is a subdivision between suicidal and unintentional intake of sodium nitrite, as well as the respective mortality and survival.

**Figure 3 ijerph-19-13996-f003:**
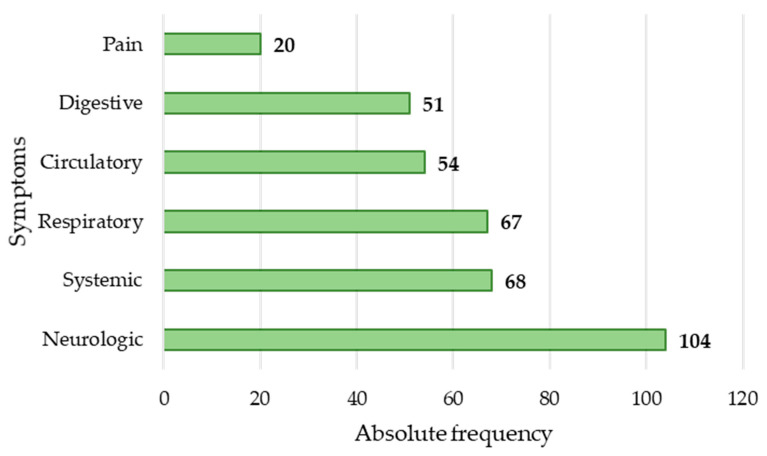
Absolute frequency of symptoms.

**Figure 4 ijerph-19-13996-f004:**
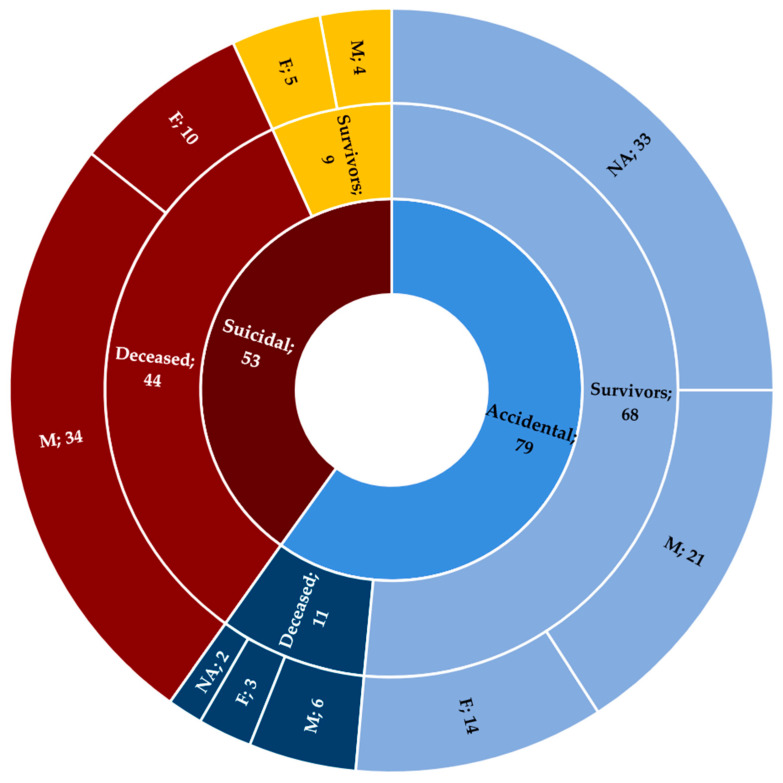
Stratification of cases in relation to intentionality and accidentality. NA: not available; M: male; F: female.

**Figure 5 ijerph-19-13996-f005:**
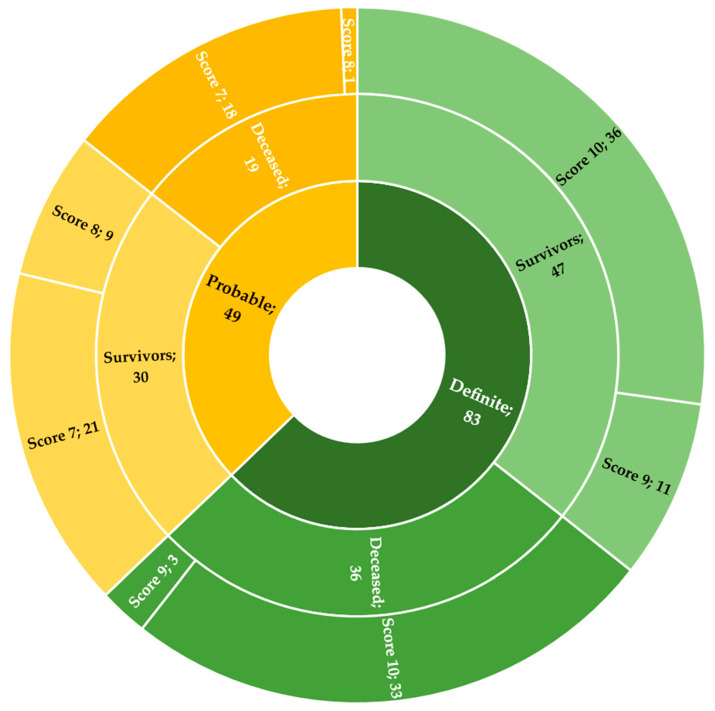
Data on the application of the Naranjo Adverse Drug Reaction Probability Scale.

**Figure 6 ijerph-19-13996-f006:**
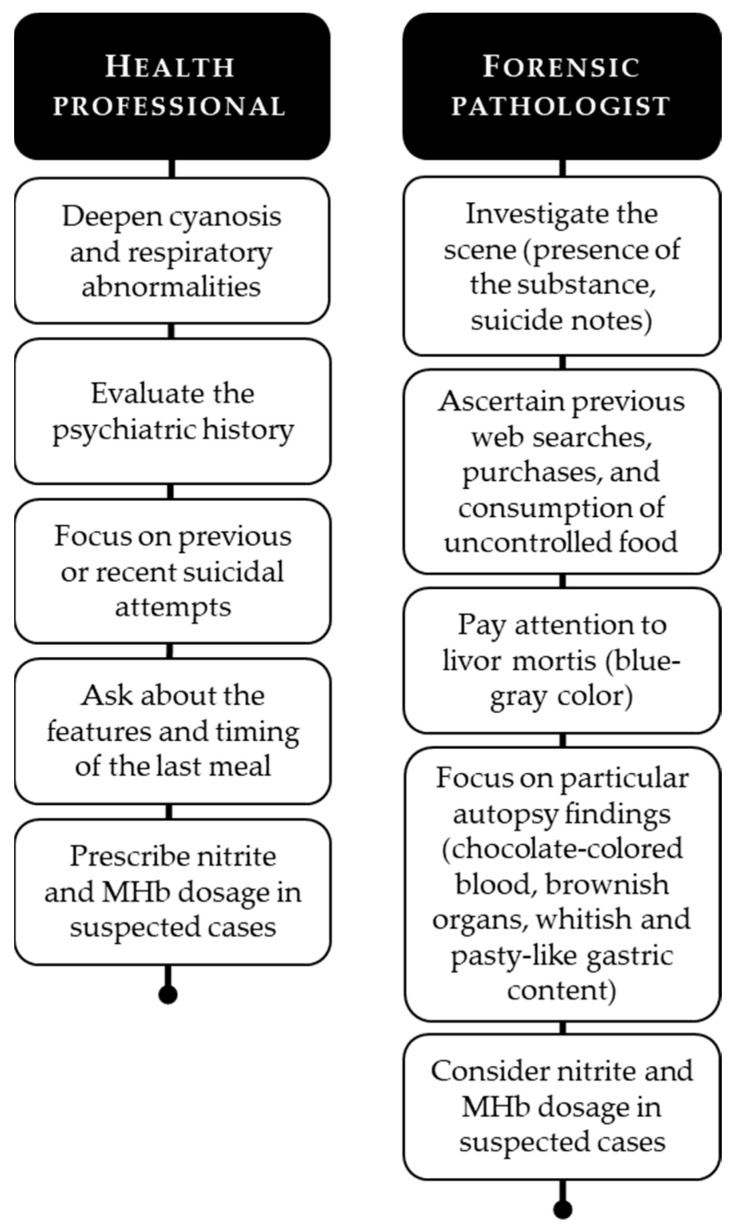
Practical advice for healthcare professionals and forensic pathologists.

**Table 1 ijerph-19-13996-t001:** Main features of the included studies.

Authors and Year of Publication	Country	Total Cases	Gender (M:F)	Age Range	Voluntary (V) or Accidental (A)	Comorbidity	Drug Association	Hospitalization	Mortality
Çağlar et al. (2016) [18]	Turkey	1	M (1)	2	A (1)	Yes (1)	No (1)	Yes (1)	No (1)
CDC MMWR (2002) [19]	USA	5	3:2	29–60	A (5)	Unknown (5)	No (5)	Yes (5)	No (5)
Cruz et al. (2018) [20]	USA	1	F (1)	23	V (1)	Yes (1)	No (1)	Yes (1)	No (1)
Cvetković et al. (2019) [21]	Serbia	3	1:2	53–70	A (3)	Yes (1) Unknown (2)	No (3)	Yes (3)	Yes (1) No (2)
Dean et al. (2021) [22]	USA	3	2:1	22–39	V (3)	Yes (1) Unknown (2)	No (2) Unknown (1)	No (3)	Yes (3)
Durão et al. (2020) [23]	Portugal	1	M (1)	37	V (1)	Yes (1)	Yes (1)	No (1)	Yes (1)
Durão et al. (2021) [24]	Portugal	1	F (1)	37	V (1)	Unknown (1)	Yes (1)	No (1)	Yes (1)
Finan et al. (1998) [25]	Ireland	3	2:1	2–4	A (3)	Unknown (3)	No (3)	Yes (3)	No (3)
Gowans et al. (1990) [26]	UK	1	F (1)	17	A (1)	Unknown (1)	No (1)	Yes (1)	Yes (1)
Harvey et al. (2010) [27]	New Zealand	1	M (1)	76	V (1)	Yes (1)	Unknown (1)	Yes (1)	Yes (1)
Hinkey et al. (2021) [28]	Canada	15	12:3	22	V (15)	Unknown (15)	Yes (8) No (5) Unknown (2)	Yes (1) No (14)	Yes (15)
Hwang et al. (2021) [29]	Korea	1	M (1)	28	V (1)	Unknown (1)	No (1)	No (1)	Yes (1)
Kaplan et al. (1990) [30]	Johannesburg	10	9:1	20–50	A (10)	Unknown (10)	No (10)	Yes (10)	Yes (1) No (9)
Katabami et al. (2016) [31]	Japan	1	M (1)	28	V (1)	Unknown (1)	No (1)	Yes (1)	No (1)
Kennedy et al. (1997) [32]	UK	3	M (3)	10–19	A (3)	No (3)	No (3)	Yes (3)	No (3)
Kim et al. (2022) [33]	Korea	10	8:2	19–35	V (10)	Unknown (10)	Unknown (10)	Yes (2) No (8)	Yes (10)
Lee et al. (2017) [34]	Korea	5	1:4	58–76	A (5)	Yes (1) Unknown (4)	No (5)	Yes (4) No (1)	Yes (1) No (4)
Lien et al. (2021) [35]	USA	1	M (1)	23	A (1)	No (1)	No (1)	Yes (1)	No (1)
Maric et al. (2008) [36]	Australia	6	1:1 Unknown (4)	Unknown (6)	A (6)	Unknown (6)	No (6)	Yes (5) No (1)	No (6)
Matin et al. (2022) [37]	USA	1	M (1)	23	V (1)	Yes (1)	Yes (1)	Yes (1)	No (1)
Matteucci et al. (2008) [38]	Italy	2	1:1	9–40	A (2)	Unknown (2)	No (2)	Yes (2)	No (2)
McCann et al. (2020) [39]	USA	5	3:2	17–35	V (5)	Unknown (5)	Yes (1) No (4)	Yes (5)	Yes (3) No (2)
Mudan et.al (2020) [40]	USA	5	3:2	16–27	V (5)	Unknown (5)	Yes (1) No (4)	Yes (5)	Yes (3) No (2)
Mun et al. (2022) [41]	Korea	2	1:1	20–26	V (2)	Unknown (2)	Unknown (2)	Yes (2)	Yes (1) No (1)
Neth et al. (2020) [42]	USA	1	F (1)	17	V (1)	Yes (1)	Unknown (1)	Yes (1)	Yes (1)
Nishiguchi et al. (2014) [43]	Japan	1	M (1)	30	V (1)	Yes (1)	No (1)	No (1)	Yes (1)
O’Neill et al. (2021) [44]	Australia	2	F (2)	31–32	A (2)	No (2)	No (2)	Yes (2)	No (2)
Padberg et al. (1939) [45]	USA	3	M (3)	55–70	A (3)	No (1) Unknown (2)	No (3)	No (3)	Yes (3)
Sajko et al. (2022) [46]	Canada	1	F (1)	15	V (1)	Yes (1)	No (1)	Yes (1)	No (1)
Sohn et al. (2013) [47]	Korea	28	8:1 Unknown (19)	48–67 Unknown (19)	A (28)	Unknown (28)	No (28)	Unknown (28)	Yes (4) No (24)
Su et al. (2012) [48]	Taiwan	2	1:1	68–72	A (2)	No (2)	No (2)	Yes (2)	No (2)
Taus et al. (2021) [49]	Italy	2	M (2)	28–33	V (2)	Yes (2)	Yes (2)	No (2)	Yes (2)
Tomsia et al. (2021) [50]	Poland	1	M (1)	23	V (1)	Unknown (1)	No (1)	No (1)	Yes (1)
Tung et al. (2006) [51]	Taiwan	1	M (1)	67	A (1)	Unknown (1)	No (1)	Unknown (1)	No (1)
Wang et al. (2013) [52]	China	3	1:2	21–68	A (3)	Unknown (3)	No (3)	Yes (3)	No (3)

M: male; F: female.

**Table 2 ijerph-19-13996-t002:** Main characteristics of the deaths stratified by the age of the subjects.

Age (Years)	Number of Cases (M:F)	Voluntary:Accidental Intake	Scene Investigation	Autopsy	Histopathology	Number of Cases with Co-Use of Drugs and/or Substances of Abuse	Co-Consumed Substance
11–20	10 (5:5)	9:1	5	7	0	2	Aripripazole (1) Citalopram (1) Bupropion (1) Clonazepam (1) Diphenhydramine (1) Midazolam (1) Trazodone (2)
21–30	22 (19:3)	22:0	10	17	2	4	Alcohol (1) Diphenhydramine (3) Quinidine (1)
31–40	8 (5:3)	8:0	5	7	2	4	Amiodarone (1) Cannabinoids (1) Citalopram (1) Codeine (1) Diazepam (1) Ibuprofen (1) Nordiazepam (1) Olanzapine (1) Quetiapine (1) Topiramate (1) Tramadol (1) Trazodone (1)
41–50	4 (4:0)	3:1	3	3	0	3	Bupropion (1) Diphenhydramine (2) Venlafaxine (1)
51–60	3 (2:1)	0:3	0	0	0	0	-
61–70	5 (4:1)	1:4	1	5	1	0	-
>70	1 (1:0)	1:0	0	0	0	0	-
NA	2	0:2	0	0	0	0	-
Total cases	55 (40:13, 2 NA)	44:11	24	39	5	13	

**Table 3 ijerph-19-13996-t003:** Summary of the post-mortem toxicology results.

		MHb (%)	Nitrite (mg/L)	Sodium Nitrite (mg/L)	Nitrate (mg/L)	Sodium (mmol/L)	Chloride (mmol/L)
Blood	Heart	9.87–83.4	1.3–170		71.69–524.8		
	Peripheral		1.1–298		83.48–476.4		
Vitreous humor			57.7			148	124
Content	Gastric	NA	6.8–16,000	13,000–24,000	5.0–137.8		
	Bowel		4.2		0.8		
Urine			24.6				
Fluid	Cerebrospinal	NA			50.5		
	Pericardial	NA	181		91.7		
Organs	Liver/Kidney mixture		0.003 mg/kg				
	Liver		0.3 mg/kg				
	Kidney		3.6 mg/kg				
Costal cartilage			3.4 mg/kg				

NA: not available.

## Data Availability

Not applicable.

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
