# Peer review of "Sodium Nitrite Intoxication and Death: Summarizing Evidence to Facilitate Diagnosis"

_ijerph, 2022, doi:10.3390/ijerph192113996_

Round 1

Reviewer 1 Report

This is a Review where authors outline an important and current issue: ingestion by sodium nitrate and his implications. 

The goal of this review is to frame the main features of deaths due to sodium nitrite intake. 

The Review is well structured and written. 

I’d like to suggest just little hides: 

1-    In the discussion between line 236 to 256 concepts should be summarized. It’s a review and it’s more important focuse on the analythics results. 

2-    I’d like to suggest to the authors to take in the account in the discussion: “Four cases of sodium nitrite suicidal ingestion: a new trend and a relevant Forensic Pathology and Toxicology challenge”.V. Bugelli, I. Tarozzi, AliceChiara ManettibF.Stefanelli, M. Di Paolo, S. Chericoni.https://doi.org/10.1016/j.legalmed.2022.102146

In this paper authors describe 4 cases, it is important to join them together to provide additional insights for discussion. 

3-    It could be better discuss what the limitations are from a medico-legal point of view, as it is important to keep in mind how thorough the inspection should always be in order to gather important elements to set the next steps.

4-    I would also like to add that in the discussion the authors in addition to focusing on preventive and social aspects could/should provide in a schematic way practical advice for the pathologist and health care professionals who will approach these cases.

Author Response

This is a Review where authors outline an important and current issue: ingestion by sodium nitrate and his implications. The goal of this review is to frame the main features of deaths due to sodium nitrite intake. The Review is well structured and written. I’d like to suggest just little hides:

  1. In the discussion between line 236 to 256 concepts should be summarized. It’s a review and it’s more important focuses on the analytics results.

Thanks for the precious suggestions. The paragraph has been summarized, focusing more attention on the results obtained.

  1. I’d like to suggest to the authors to take in the account in the discussion: “Four cases of sodium nitrite suicidal ingestion: a new trend and a relevant Forensic Pathology and Toxicology challenge”. Bugelli, I. Tarozzi, Alice Chiara Manetti, F. Stefanelli, M. Di Paolo, S. Chericoni. https://doi.org/10.1016/j.legalmed.2022.102146. In this paper authors describe 4 cases, it is important to join them together to provide additional insights for discussion.

Thanks for the valuable suggestion. The recommended reference has been considered and has been added.

  1. It could be better discuss what the limitations are from a medico-legal point of view, as it is important to keep in mind how thorough the inspection should always be in order to gather important elements to set the next steps.

Thanks for the precious suggestions. The recommended considerations have been deepened and dealt with in the discussion.

  1. I would also like to add that in the discussion the authors in addition to focusing on preventive and social aspects could/should provide in a schematic way practical advice for the pathologist and health care professionals who will approach these cases.

Thanks. The discussions were implemented by developing this theme, also by adding Figure 6.

Reviewer 2 Report

This is a good review of the literature compiling recent reports about sodium nitrite toxicity and death.  

The language is at times confusing, and would benefit from editing by a native speaker of English

I wonder about some of your references as they don't appear appropriate (eg. 74, 75 in particular)

I think stating that internet censoring is required by authorities is a charged statement that doesn't belong in a scientific paper:

"Given the spread of the phenomenon, it would be advisable, on the one hand, to implement tight checks of online content by the authorities and, on the other hand, to tackle the health problem"

The discussion largely focuses on suicidal sodium nitrite ingestion, but does not go over the significant problem of accidental ingestion to a great degree.  The paper would benefit from a more well-rounded discussion including lessons learned about clinical assessment in the living and circumstances of accidental ingestion (and how to avoid them).

Author Response

This is a good review of the literature compiling recent reports about sodium nitrite toxicity and death.

  1. The language is at times confusing, and would benefit from editing by a native speaker of English.

Thanks for the precious suggestions. The English language has been revised by a native speaker.

  1. I wonder about some of your references as they don't appear appropriate (eg. 74, 75 in particular).

Thanks. The previous references have been checked and no. 74 and 75 were eliminated.

  1. I think stating that internet censoring is required by authorities is a charged statement that doesn't belong in a scientific paper: “Given the spread of the phenomenon, it would be advisable, on the one hand, to implement tight checks of online content by the authorities and, on the other hand, to tackle the health problem”.

Thanks for the valuable suggestions. The authors did not intend to express in terms of censorship but to the protection of the categories at risk from exposure to content suggesting suicide. The section indicated by the reviewer has been supplemented with specifications on the advisability of checking internet content to protect public health.

  1. The discussion largely focuses on suicidal sodium nitrite ingestion, but does not go over the significant problem of accidental ingestion to a great degree. The paper would benefit from a more well-rounded discussion including lessons learned about clinical assessment in the living and circumstances of accidental ingestion (and how to avoid them).

We are very grateful to the reviewer for this comment. A paragraph on accidental ingestions as well as on the epidemiological and clinical implications of the same has been added in the "Discussion" section.